# Nanoparticles as a Promising Strategy to Mitigate Biotic Stress in Agriculture

**DOI:** 10.3390/antibiotics12020338

**Published:** 2023-02-06

**Authors:** Gonzalo Tortella, Olga Rubilar, Joana C. Pieretti, Paola Fincheira, Bianca de Melo Santana, Martín A. Fernández-Baldo, Adalberto Benavides-Mendoza, Amedea B. Seabra

**Affiliations:** 1Centro de Excelencia en Investigación Biotecnológica Aplicada al Medio Ambiente (CIBAMA), Facultad de Ingeniería y Ciencias, Universidad de La Frontera, Av. Francisco Salazar 01145, Temuco 4811230, Chile; 2Departamento de Ingeniería Química, Facultad de Ingeniería y Ciencias, Universidad de La Frontera, Av. Francisco Salazar 01145, Temuco 4811230, Chile; 3Center for Natural and Human Sciences, Federal University of ABC (UFABC), Avenida dos Estados, Saint Andrew 09210-580, Brazil; 4Instituto de Química San Luis (INQUISAL), Departamento de Química, Universidad Nacional de San Luis, CONICET, Chacabuco 917, San Luis D5700BWS, Argentina; 5Department of Horticulture, Universidad Autónoma Agraria Antonio Narro, Saltillo 25315, Mexico

**Keywords:** nanoparticles, biostimulants, biotic stress, crops, antimicrobial, agriculture

## Abstract

Nanoparticles are recognized due to their particular physical and chemical properties, which are conferred due to their size, in the range of nanometers. Nanoparticles are recognized for their application in medicine, electronics, and the textile industry, among others, but also in agriculture. The application of nanoparticles as nanofertilizers and biostimulants can help improve growth and crop productivity, and it has therefore been mentioned as an essential tool to control the adverse effects of abiotic stress. However, nanoparticles have also been noted for their exceptional antimicrobial properties. Therefore, this work reviews the state of the art of different nanoparticles that have shown the capacity to control biotic stress in plants. In this regard, metal and metal oxide nanoparticles, polymeric nanoparticles, and others, such as silica nanoparticles, have been described. Moreover, uptake and translocation are covered. Finally, future remarks about the studies on nanoparticles and their beneficial role in biotic stress management are made.

## 1. Introduction

In recent decades, the population has increased notably, and the pressure on food production has grown exponentially. In this sense, food production has also significantly increased the number of applied pesticides in the environment and soil pollution [1]. In addition, the excessive use of pesticides has developed resistance in microorganisms, making them more challenging to control and affecting productivity [2]. On the other hand, the effects of global climate change have reduced the cultivable surface due to erosion processes [3]. Drought, salinity, and high temperatures, among other factors triggered by changing climatic conditions, have also increased losses in agricultural production due to biotic stress [4]. All these problems have forced us to search for alternative solutions to combat the effect of biotic stress and increase food production and quality. Among the alternatives available is the agricultural use of nanomaterials (NMs) [5,6].

In recent years, the use of NMs has gained importance in several areas, such as medicine, cosmetics, electronics, communications, energy production, textiles, agriculture, and food processing [5]. NMs are defined as structures, aggregates, or agglomerates with at least one external dimension less than 100 nm [7] or with a volume-specific surface area (VSSA) > 60 m^2^ cm^−3^ [8]. The large surface area relative to the volume is the property that essentially defines the biostimulant capacity of NMs [9] and the differences in their physicochemical behavior compared to that of bulk materials [10].

Different ways of classifying NMs have been used. Classification is based on properties such as dimensions, origin (natural or synthetic), chemical composition, toxicity nature, and homogeneity (with one or more components) [10,11]. One common way to classify NMs is to consider the number of dimensions outside the nanoscale range. There are 0D NMs [nanoparticles with all dimensions ≤ 100 nm], 1D (e.g., nanofibers, nanowires, nanotubes, nanorods with one dimension > 100 nm), 2D (e.g., nanolayers, graphene, nanocoatings, nanofilms with two dimensions > 100 nm), and 3D (zeolites and other porous materials, powders and dispersions with three dimensions > 100 nm) [12].

Nanoparticles have been mentioned as a potential tool to alleviate the damage caused by abiotic stress. Metal nanoparticles have shown many applications in plants. Silica nanoparticles have been shown to promote plant growth and induce plant resistance against biotic stress, as reviewed by [13]. In this sense, copper, zinc oxide, and selenium nanoparticles have demonstrated excellent results when used as nanofertilizers [14,15,16]. Nanoparticles have also shown the ability to be used as inducers of the biosynthesis of phytohormones, regulating plant growth and metabolism under abiotic stress [17,18]. In this regard, nitric oxide-releasing chitosan nanoparticles are an efficient tool against the adverse effects caused by saline stress [19]. Soil treatment with nitric oxide-releasing chitosan nanoparticles has demonstrated that it protects the root system and promotes the growth of soybean plants under copper stress [20].

Nanoparticles in pest management have been revolutionary for agriculture because they facilitate a substantial decrease in pesticide use. At the same time, several nanoparticles, such as polymeric, metal, and metal-oxide nanoparticles, have shown high efficiency in treating biotic stress in crops [17]. In this sense, silver and copper nanoparticles are the most studied nanoparticles due to their high capacity to act as antimicrobial compounds [21,22]. However, due to their antimicrobial ability, biocompatibility, and biodegradability (due to their nontoxic nature), chitosan nanoparticles have played a significant role in biotic stress control studies in plants [23,24]. Polymeric nanoparticles are recognized not only by their antimicrobial capacity but also by their capacity to be used as carriers (nanoencapsulation) of biocontrol agents and to alleviate plant biotic stress [25,26]. In recent work, it has also been demonstrated that nanoparticles can act by modulating plant metabolic pathways, allowing the amelioration of biotic stress in plants [27]. Therefore, in this context, the present review considers the current and relevant information findings related to the use of polymeric and metal or metal oxide nanoparticles to combat biotic stress in plants. In addition, future guidelines in the area are also provided.

## 2. Uptake and Translocation of Nanoparticles in Plants

The application of nanoparticles on plants has been widely reported to control bacteria, fungi, nematodes, and insects, among others. Nevertheless, the interaction of nanoparticles with the plant system (Figure 1) constitutes a complex process at the root and foliar levels [28]. The unique properties of nanoparticles, such as large surface area and high reactivity, allow them to easily interact with vegetable tissue. Furthermore, nanoparticle size, concentration, stability, and chemical configuration play an essential role in uptake and translocation inside plants [29]. Chemically, the mobility and adherence of nanoparticles into plant tissue depend on gravity, Brownian motion, double layer forces, and van der Waals forces, as reviewed by [30]. Nanoparticles can penetrate the plant system through the aerial pathway by structures such as the hydathode, stomata, and trichomes or by wounds produced by phytopathogens [31].

Moreover, the root system constitutes an essential and complex pathway of nanoparticle uptake due to its interaction with the soil [30,32]. Consequently, the ability of nanoparticles to mitigate biotic stress in plants strongly depends on their uptake and translocation within the plant system [13]. However, another essential factor to consider in this interaction is the plant species and the growth stage in which it is found. Each plant species has specific barriers that regulate the entry of nanoparticles via anatomical aspects (i.e., the composition of the cell wall governs the passage of nanoparticles according to their solubility and chemical nature) [33]. Furthermore, the morphology and chemical structures of leaves and roots play an essential role in the uptake and translocation of nanoparticles into the plant system. Once the nanoparticles enter the plant system, they can modulate morphological, biochemical, and physiological properties to improve biotic stress tolerance.

The rhizosphere is a narrow dynamic zone influenced by a complex interaction between soil microorganisms and root exudates [34]. In the first instance, plant root uptake of nanoparticles is strongly influenced by rhizospheric conditions, root exudates, and root morphology [35]. In addition, root exudates are considered beneficial phenomena of the root system that control the chemical and physical properties of the rhizosphere [36]. Therefore, these factors strongly influence the uptake of nanoparticles, producing their adsorption, immobilization, chemical transformation, aggregation, speciation, dissolution, or interaction with organic matter. Otherwise, the surface charge of roots directly affects the uptake and translocation of nanoparticles, directly by the secretion of exudates and mucilage from the root hairs [31]. Specifically, the mucilage layer confers a negative charge to root secretions, which is an essential factor in the adherence of nanoparticles on the surface.

The morphology of roots is one of the main parameters to consider in evaluating the effect of nanoparticles. In the first stage, nanoparticles are adsorbed on the root surface, interacting with mucilage and other compounds secreted by the root [37]. Once the nanoparticles enter the root, they must interact with different root structures, such as the epidermis, cortex, Casparian strips, and endodermis. Once the nanoparticles enter the epidermis, they are translocated throughout the plant via the apoplastic or symplastic pathway [38]. According to various authors’ reports, nanoparticles can be translocated through the apoplastic path, where they enter plant tissue through cell wall pores and diffuse into the intracellular space between the cell membrane and cell wall [39]. According to recent reviews, the pores of the cell walls may increase in size when they are exposed to nanoparticles, allowing their entry. Another route of entry is through the intercellular space generated by damage to the root tissue [38]. The transport of nanoparticles through the apoplastic pathway implies that the Casparian strips restrict their passage through the cell wall, cell membrane, and cortex due to their lipophilic nature. However, nanoparticles can avoid the Casparian strips and enter the vascular system by the apoplastic pathway [29].

On the other hand, some authors have reported the translocation of nanoparticles through the symplastic pathway, where nanoparticles enter the cell membrane and cytoplasm or adjacent cell wall pores known as plasmodesmata. According to the review of [38], nanoparticles can cross the cell membrane by aquaporins, membrane channels, and endocytosis. It was reported that metal nanoparticles are translocated through endocytosis, but that this process strongly depends on the physicochemical properties of the nanoparticle surface. In general, the endocytosis process can be carried out for nanoparticles between 5 and 15 nm, which is an important restriction parameter to uptake and translocation. However, the process carried out through the receptor-mediated clathrin-dependent fluid-phase endocytosis allows the translocation of NPs in the range of 70 to 120 nm [38].

Leaf status is essential in nanoparticle translocation into the plant system. For example, the anatomical structure and biochemical composition of young and senescent leaves determine the uptake and translocation of nanoparticles [37]. In addition, symptoms of necrosis and damage in the leaf surface can facilitate the entry of nanoparticles into the plant, such as the attack of pathogens and diseases. Nanoparticles can enter by foliar exposure via the cuticular pathway and stomatal routes. Leaves are covered by a waxy cuticle layer, constituting the first barrier to the nanoparticles entering the plant. The cuticle layer protects plant leaves against water loss and regulates the exchange of solutes [40]. The cuticle has two pathways to uptake depending on its lipophilic or hydrophilic nature. It was reported that nanoparticles up to 5 nm in size could enter directly through the cuticle. It is still under investigation whether nanoparticles with a larger size can diffuse through this structure [38]. Thus, biochemical or structural changes in the cuticle by environmental or biotic factors can modify the uptake and translocation of nanoparticles.

Stomata are tiny pores that regulate the interchange of CO_2_ and water vapor between plants and the environment [41]. Therefore, the stomatal pathway can play a relevant role in the uptake and translocation of nanoparticles into plants through the phloem system, despite the few studies that show it. It has been indicated that the morphological size pore of stomata has a length of 25 µm and a width from 3 to 10 µm [38]. From this, it is suggested that stomata can transfer nanoparticles inside plants by a size-dependent process. Nanoparticles can accumulate in the stomata and later be translocated by an up–down method through the phloem. Furthermore, hydathodes, characterized by tiny pores found in the leaf tip in angiosperm plants, are another structure through which nanoparticles can enter, according to what has been reported. Hydathodes play an essential role in decreasing excess water through the guttation process. Otherwise, microorganisms in the phyllosphere can regulate the entry of nanoparticles into the plant through the secretion of metabolites, which improves or prevents translocation [29].

## 3. Potential Adverse Effects of Nanoparticles on Plants

Despite the positive effects found in plants after exposure to nanoparticles [42,43], metal or metal oxide nanoparticles such as copper, copper oxide, zinc oxide, silver, or titanium oxide nanoparticles have demonstrated adverse or contradictory effects on plants [44]. Induced stress due to the presence of copper nanoparticles on *Oryza sativa* caused a reduction in photosynthetic rate, a low number of thylakoids per granum, and decrease in transpiration rate and stomatal conductance [45]. Similar results were reported in *Lactuca sativa* and *Daucus carota* due to copper oxide nanoparticles [46]. All concentrations between 0.8 and 798.9 mg L^−1^ caused an increase in root diameter in both plants. However, decreases in root length and germination rate were evidenced as the nanoparticle concentration increased. Phytotoxicity has also been determined in the case of zinc oxide nanoparticles. Exposure to 100 and 1000 mg L^−1^ of zinc nanoparticles on *Salicornia persica* plants caused a decrease in shoot length by more than 50% compared to non-treated plants. The damage was caused by ROS generation and lipid peroxidation, which was three times higher than for non-treated plants [47]. Similar results were found in *Cajanus cajan* L. seeds expoded to 50, 100, 150, 200, 250 mg L^−1^ of zinc nanoparticles [48]. The authors reported that 200 and 250 mg L^−1^ caused a reduction in the % of seed, number of leaves, shoot length, root length, width of leaves, and fresh and dry weight of plants [48]. It is important to mention that the damage of metal or metal oxide nanoparticles is governed by soil pH and/or plant species, which influence the Zn availability and phytotoxicity of zinc nanoparticles [49]. The effects of zinc nanoparticles on calcareous soil (alkaline pH) compared with acidic soil were less evident due to the low availability of zinc in alkaline soils. However, depending on the species, the damage can be more pronounced [49]. Phytotoxicity and cytotoxicity in silver nanoparticles has also been reported [50,51,52,53]. Biogenic silver nanoparticles synthesized by *Aloe vera* extract at 1 and 3 mM proved to be harmful to *Brassica* sp. seedlings in hydroponical cultures [54]. Nanoparticles caused severe alterations in photosynthesis and induced oxidative stress by ROS generation causing DNA degradation and cell death. Antioxidant enzymes (ascorbate peroxidase and catalase) were also inhibited. However, interestingly the damage produced by silver nanoparticles was less compared with that of AgNO_3_ at the same concentrations [54]. In a recent work [55] demonstrated that silver nanoparticles with different surface properties display different inhibition grades on the growth of monocots and dicots model plants. The different silver nanoparticles (15  ±  3 nm) were synthesized using trisodium citrate, tannic acid, and cysteamine hydrochloride, leading in nanoparticles with different surface charges (positive or negative). The silver nanoparticles caused damage at the root or shoot level in monocots and dicots model plants. However, the injury to plants was more significant with positively charged nanoparticles and silver ions from AgNO_3_ [55]. Damage produced by titanium oxide nanoparticles has also been reported, with similar effects produced by other metal or metal oxide nanoparticles [44]. However, the results of phytotoxicity for titanium nanoparticles have shown that these nanoparticles caused less damage on plants than other metal or metal oxide nanoparticles. Inhibition of leaf growth and alteration in the root water transport system [56], growth inhibition and damage to root cell membranes [57], or ROS generation and inhibition of chlorophyll synthesis [58] have been reported, although generally at high concentrations of nanoparticles, demonstrating that the use of titanium nanoparticles on plants could be safer from a phytotoxicity point of view. In a recent work, it is also demonstrated that although titanium nanoparticles can cause phytotoxic effects in plants, hermetic effects are also revealed [58]. At 100 mg L^−1^, the elongation of shoots and roots and total biomass growth were significantly promoted by nanoparticles as well as the proline content. However, over 1000 mg L^−1^ a clear inhibition in these areas was detected. The use of nanoparticles in plants has shown various beneficial effects. However, it is clear that the dosage must be considered depending on the crop type and soil type, among other factors, to avoid damage and non-desirable effects on the non-target organism.

## 4. Potential Use of Polymeric Nanoparticles

Different types of nanomaterials can be employed in the field of agriculture, highlighting metal or metal oxide NPs, such as silver NPs (AgNPs) and copper oxide NPs (CuO_2_) NPs, as well as lipid or polymer-based NPs, such as micelles and chitosan NPs (CS NPs) [59]. When comparing the two main classes of NPs (organic and inorganic NPs), there is the key difference concerning applications focusing on biotic stress: (i) Inorganic NPs usually demonstrate intrinsic activity against the pathogen (e.g., AgNPs directly demonstrate antifungal and antibacterial activity against plant-infecting nematodes, bacteria and fungi) [60]. (ii) Polymeric NPs are mainly employed as nanocarriers, promoting the efficient release of the active agent [61]. Polymeric NPs have been extensively studied in recent years for their potential use in the controlled release and protection of active compounds against unfavorable environmental conditions. Their high stability and ability to release active compounds in a specific zone of plant target of polymeric NPs have led to great interest in their application in agriculture. Furthermore, their biodegradability and biocompatibility means that polymeric NPs are characterized by low toxicity. In addition, these NPs have the capacity to encapsulate a high number of active compounds with low environmental impact due to their slow release [62]. Therefore, polymeric NPs commonly require combining with other active molecules, such as antibiotics, pesticides, herbicides, or even micronutrients, to achieve their functionality [60].

Among different polymeric nanostructures, nanomicelles, nanocapsules, and nanospheres are synthesized and employed in various fields [63]. Despite the definition of nanomaterials according to European Union Law, polymeric NPs may demonstrate small sizes (until 100 nm) or larger sizes (from 100 nm to 1000 nm) and still display unique properties [63]. Overall, polymeric NPs in agriculture can efficiently deliver the loaded molecule employing lower amounts of the active agent, promoting extended adhesion and uptake, enhancing thermal and photostability, and ameliorating soil leaching [59]. Considering these points, commonly developed polymeric NPs for agricultural application are preferably composed of biocompatible and biodegradable polymers with low toxicity and cost. A schematic representation of polymeric NPs in agriculture is shown in Figure 2.

Among polymeric nanomaterials for agricultural applications, chitosan is the most commonly used biopolymer due to its biocompatibility, biodegradability, and relatively low cost [25]. In plants, chitosan stimulates plant growth and induces tolerance to (a)biotic stresses [64]. Interestingly, chitosan is involved in the modulation of second messengers, such as nitric oxide (NO), Ca^2+^, reactive oxygen species (ROS), and phytohormones, regulating several plant responses upon chitosan treatment [65,66]. Chitosan is reported to induce systemic resistance in plants due to its action as an effective biotic elicitor [67]. Chitosan administration to plants is reported to increase plant tolerance to a wide range of pathogens [68]; moreover, chitosan nanoparticles can be used to load active molecules, such as NO donors, to enhance plant growth [69]. Chitosan and chitosan NPs (empty NPs) (0.1–5.0 mg/mL) were evaluated in the control of *Fusarium andiyazi* in wilt disease in tomato (*Solanum lycopersicum*) [70]. The maximum tested concentration of both chitosan and chitosan NPs led to the maximum radial mycelial growth inhibition (by 55 and 74%, respectively). In fact, chitosan and chitosan NPs inhibited *Fusarium andiyazi* development in tomatoes. They acted as effective plant defense elicitors, and superior effects were observed for the nanoform of chitosan compared to bulk chitosan [70]. This result is expected because nanomaterials, which have a larger surface area and charge density, increase their adsorption by fungal cells and promote the leakage of cellular components of pathogen cells, causing cell death [71]. Deposition of chitosan around the plant sites where pathogens penetrate creates a physical barrier, preventing pathogen uptake and colonization in plants. Moreover, chitosan stimulates ROS generation and the accumulation of phenolic compounds that promote lignification and inhibit the action of proteinase [72].

Furthermore, as stated before, chitosan NPs can also be used as nanocarriers of traditional agrochemicals, enhancing their effectiveness with fewer side effects. For instance, chitosan NPs (300 nm) were loaded with paraquat, a fast-acting herbicide, and used more safely to control weeds in agriculture [73]. Recently, chitosan thiamine NPs were used to activate defense responses caused by *Fusarium oxysporum* f. sp. Cicero in chickpeas [74]. An increase in nonenzymatic and enzymatic antioxidants and higher lignin deposition in vascular bundles of chickpea steam tissues were reported compared to the control group. These results correlate with plant resistance against wilt pathogens [74]. Plant defense against biotic stress is permeated by ROS generation and enhanced activity of antioxidant enzymes such as peroxidase, superoxide dismutase, catalase, and glutathione peroxidase, and antioxidant biomolecules such as flavonoids [75]. In an exciting approach, chitosan NPs and salicylic acid were sprayed before and after *Puccinia striiformis* (an obligate fungal parasite) inoculation in wheat leaves to mitigate leaf rust disease [76]. This work aimed to propose an alternative approach to the use of fungicides. Chitosan NPs increased the incubation and latent period and decreased the infection type, number, and size of pustules. Salicylic acid was also practical but less effective than chitosan NPs.

It should be noted that the authors evaluated the effects of empty chitosan NPs and pure salicylic acid. The result of salicylic acid encapsulated in chitosan NPs should be investigated. Similarly, the essential oil peppermint oil was encapsulated into chitosan NPs (563 nm, encapsulation efficiency of 64%) and successfully used to promote stored food pest control scheduled for the pest insets *Sitophilus oryzae* and *Tribolium castaneum* [77]. In addition to chitosan, the natural polymer poly(ε-caprolactone) (PCL) was successfully used to prepare nanocapsules containing the herbicide atrazine against *Bidens pilosa* (weed species) and its effect on soybean plants [78]. PCL NPs containing atrazine (483 nm) were also applied to control weeds with the target (*Brassica* sp.) and nontarget *Zea may* [79]. Encapsulation of the herbicide into PCL NPs reduced its mobility in the soil and reduced atrazine genotoxicity. Atrazine-containing PCL was effective in the control of agricultural weeds, whereas it reduced the toxicity of the herbicide [80]. Recently, commercial herbicides (fenoxaprop-P-ethyl, tribenuron-methyl and metribuzin) were incorporated into degradable polymeric microparticles of polyhydroxyalkanoates (PHAs) of two types—poly-3-hydroxybutyrate [P(3HB)] and poly(3-hydroxybutyrate-co-3-hydroxyvalerate [P(3HB/3HV)] [81]. The encapsulation efficiency was found to be 24–48%, which should be improved. The microparticles showed a sustained release of the herbicides over 30 days and effectiveness against *Elsholtzia ciliata* weed plants [80]. A polymeric nanodelivery system was obtained with star polymer-based cyantraniliprole. Its toxicity was demonstrated against the pest *Frankliniella occidentalis* (WFT, an insect pest) and the predator *Orius sauteri* [81]. The nanodelivery system was effective and selective in pest control.

Other intelligent and exciting strategies have recently been used to create new and efficient nanocarriers to mitigate biotic stress in agriculture. In light of sustainability, renewable plant oil-based polymers were prepared to deliver pesticides (a model pesticide Azox) [82]. Alginate-based NPs have also been used to promote the sustainable release of agrochemicals [83]. In summary, biopolymers have been used as nanocarriers to encapsulate agrochemicals to mitigate biotic stress in crop production. Table 1 brings together some polymeric NPs and their effects on biotic stress in crops.

## 5. Potential Uses of Metal and Metal Oxide Nanoparticles

Metal-based NPs have been reported as promising materials to control plant pests and diseases, in addition to improving plant growth and vigor under different stress conditions. Therefore, these NPs enhance plant biomass and crop production yield through diverse mechanisms, including crop protection (nano pesticides), stress tolerance, soil enhancement, and crop growth [84]. These NPs can have an effect against the pathogens themselves or improve defense against diseases by enhancing plant nutrition, suppressing pathogen infections (bacterial, fungal, viral), and directly increasing nutrition quality, as well as crop yield [85]. They ameliorate the stress response mainly by inducing the regulation of the plant antioxidant systems and endogenous plant hormones and act on the transcriptional regulation of stress-related genes, which summarizes reactive oxygen species (ROS) suppression in the plant and favors plant growth and development [17]. Regarding direct action against pathogens, there are different antimicrobial mechanisms performed by metal NPs. They can cause DNA damage, cell membrane damage, and interruption of electron transport through ion release and internalization. They can also generate ROS, which can cause enzyme disruptions, protein denaturation, DNA damage, etc. [86]. Despite the beneficial effects of metal-based NPs on agriculture, some researches have evaluated their toxicological impact on human health and ecosystems. Some physico-chemical properties of metal-based NPs such as size, stability, and shape have an essential role in determining the toxicological effects of metal-based NPs. Nevertheless, the advantages and important effects of this type of NPs have resulted in studies of dose-response to determine the concentration and specific amount needed to produce beneficial effects without producing adverse side effects [87]. Table 2 brings together some metal-based NPs and their effects on biotic stress in crops.

Herein, we highlight the uses of Ag and Cu NPs. Ag NPs were reported as strong nanopesticides against several phytopathogens, such as *Alternaria alternata*, *Pyricularia oryzae*, *Sclerotinia sclerotiorum*, *Fusarium oxysporum*, and *Cladosporium cucumerinum* [84,85,86,87,88,89,90,91,92,93]. Some of their mechanisms of action include ion release, induction of pits and gaps in the bacterial membrane, and interaction with disulfide or sulfhydryl groups of enzymes that lead to disruption of metabolic processes [86,94]. In addition to their antimicrobial activity, Ag NPs also increased seed germination and modified the biochemical profile of *Silybum marianum*, increasing the total content of phenols, flavonoids, protein content, peroxidase activity and superoxide dismutase activity [95]. In another study, Ag-priming of cabbage seeds enhanced cabbage seed germination speed, seedling growth, and yield. Additionally, the contents of Fe and several essential amino acids in cabbage leaves were increased several-fold by AgNP seed priming, increasing the plant’s nutritional value [96]. Cu NPs are also potent nano pesticides and enhancers of plant growth and nutrition. Their antimicrobial activity involves crossing nanoparticles from the bacterial cell membrane and damaging vital enzymes [86]. They have antimicrobial activity against important phytopathogens, such as *Phoma destructiva*, *Curvularia lunata*, *Alternaria alternata*, *Fusarium oxysporum*, *Clavibacter michiganensis*, *Gibberella fujikuroi*, *Rhizoctonia solani*, *Xanthomonas axonopodis*, *Aspergillus niger*, *Colletotrichum gloeosporioides*, and *Drechslera sorghicola* [17,97,98]. The bifunctional role of Cu NPs as nano pesticides and plant growth promoters in tomatoes was reported by Lopez-Lima and collaborators. The foliar application of Cu NPs reduced Fusarium wilt incidence and severity by 68 and 66.5%, respectively, and increased growth and chlorophyll content [43]. In another report, the foliar application of Cu NPs increased the fruit quality of tomatoes by inducing the accumulation of bioactive compounds such as vitamin C, lycopene, total phenols, and flavonoids [99].

In addition to these two NPs, another metal-based nanoparticles such as, TiO_2_ NP, shows promising applications for mitigating biotic stress in crops. They can cause oxidative stress via ROS generation and lipid peroxidation, leading to enhanced membrane fluidity and disruption of pathogen cell integrity [86]. Biogenic TiO_2_ NPs synthesized by *Chenopodium quinoa* leaf extracts showed antimicrobial activity against *Ustilago tritici*, responsible for causing wheat rust, inhibiting up to 75% of mycelial growth [100]. Moreover, Satti and collaborators showed that TiO_2_ NPs synthesized by using *Moringa oleifera* leaf aqueous extract have antimicrobial activity against *Bipolaris sorokiniana*, which causes spot blotch disease in wheat plants, and stabilize the plant’s relative water content, membrane stability index, chlorophyll content, and soluble sugar, protein, proline, flavonoid, and phenolic contents to induce disease tolerance in wheat plants [101].

ZnO NPs are another metallic NP that should be noted. They can be internalized into pathogen cells, generate ROS on the surface of the particles, cause membrane dysfunction, and release zinc ions [86]. In this sense, photoactivated ZnO NPs inhibited the growth of *Botrytis cinerea*, the cause of gray mold in strawberries, by 80%. Spraying ZnO NPs on strawberries reduced *Botrytis cinerea* incidence by 43%, enhanced crop production by 28.5%, and stopped the spoilage of harvested fruits during storage by 8 days [102]. In a different approach, the inoculation of ZnO NPs in the soil, which was synthesized through *Matricaria chamomilla*, decreased the *Ralstonia solanacearum* population responsible for causing bacterial wilt in tomatoes and disease severity, as well as improving plant growth. The ffected bacterial cells showed morphological deformation, such as disruption of the cell membrane and wall and leakage of cell contents, probably because of the release of Zn^+2^ ions [103].

It is essential to note that the responses are dose-dependent and that NPs can cause phytotoxicity to crops if not used correctly. They can have severe effects on seed germination, plant biomass, apical growth, and photosynthetic efficiency. They can directly affect plant homeostasis through ion release by NPs, which causes DNA damage by binding with DNA bases, or have indirect effects through ROS production, changing the activity of antioxidant enzymes. The alteration of plant cells’ redox balance can result in the accumulation of free radicals, causing modifications in cell signaling mechanisms and oxidative damage to biomolecules [104].

## 6. Other Nanoparticles

In addition to metal-based NPs and polymeric NPs, other important NPs have been successfully tested in agricultural applications to alleviate plant stress. Nanosilica (or SiO_2_ NPs) has emerged as a critical player in crop production and protection under biotic and abiotic stresses. The broad biological use of nanosilica results from its biocompatibility properties and a high surface-to-volume ratio [105]. Nanosilica demonstrates superior effects in plants compared to bulk silica due to its size at the nanoscale, allowing its fast uptake by the apoplastic pathway and translocation in plant tissues [106], and acts as nanocarriers for active molecules [107].

The beneficial impact of silica nanoparticles on promoting tolerance in plants against stress is well documented. However, the number of experimental studies and reviews focusing on abiotic stress seems higher than those on biotic stress. On the other hand, in a recent study, Fan et al. [108] reported that, although the favorable impact of nanosilicon is well documented, the effect depends on different factors, such as the plant species used as a biological model, the class of nanoparticles used, the type of application (generally foliar or to the substrate), and the concentration and bioavailability of nanosilicon. These findings indicate the need for more studies to define schemes for using silica nanoparticles on a commercial scale.

Silica NPs have great agricultural potential by directly impacting plant growth [108,109,110,111]. Silica-based nanomaterials increase the uptake and translocation of silica, which in turn reduces the generation and accumulation of reactive oxygen species (ROS) and lipid peroxidation, conferring greater tolerance to biotic and abiotic stress [106,107,112].

Empty silica NPs can be used as biostimulants, nano fertilizers, herbicides, and pesticides and act as nanocarriers for nucleotides, proteins, or other active molecules in agriculture [113]. Nanosilica decreases the entry of heavy metals and sodium ions into plants, alleviating salinity and heavy metal toxicity. Notably, nanosilica deposition in plant leaves increases plant defense against pathogens. Nanosilica has antibacterial and antifungal properties in addition to the ability to biostimulate plant cells. Biostimulation enhances plant defense by increasing levels of phenolics and activity of antioxidant enzymes in plants, improving plant resistance against pathogens [106].

Moreover, nanosilica is known to induce the expression of defense genes [33,114]. Indeed, nanosilica can cause a plant immune response (acquired resistance), strongly contributing to plant defense under biotic stress [112]. This cascade signaling pathway involves nitric oxide (NO), a key molecule in plant growth and protection under (a)biotic stress conditions [69], and the phytohormone salicylic acid, which activates pathogenesis-related genes in plant cells [112].

Silicon-based nanomaterials have been prepared and used as agents and nanocarriers of pesticides in crop protection against pathogens, as reviewed by Zhang et al. [115]. Recently, spherical silicon NPs (size of 45 nm and negative zeta potential of −26 mV) synthesized by the biogenic route (from *Fusarium oxysporum* SM5) demonstrated nematicide effects caused by *Meloidogyne incognita* in eggplant [111]. Albalawi et al. [116] carried out the biosynthesis of Silica nanoparticles using *Aspergillus niger*. The silica nanoparticles showed in vitro antifungal activity against *Alternaria solani* and substantially decreased the damage of *A. solani* when sprayed (100 mg L^−1^) on eggplants; additionally, a substantial improvement was observed in the concentration of antioxidant metabolites and enzymes. Similarly, Wang et al. [114] described the impact of foliar spraying of silica nanoparticles (650 mg L^−1^) on the bacterial wilt caused by *Ralstonia solanacearum* in tomatoes. The authors observed a significant decrease in infection damage indices, an increase in antioxidant metabolism, and a more remarkable synthesis of compounds associated with defense against biotic stress.

On the other hand, studies revealed that silicon NPs inhibited egg hatching, and the percentage of mortality of second-stage juveniles of root-knot nematodes ranged from 87% to 98.5% after 72 h of exposure to NPs at 100 and 200 ppm. Interestingly, combining silicon NPs (at 100 ppm) with commercial nematicides at their half-recommended doses further inhibited egg hatching and second-stage juvenile root-knot nematode mortality. These results suggest that silicon NPs can be administered with traditional nematicides for pathogen control in crop production [111]. In another study [117], it was shown that the susceptibility of tomato plants against the Root-Knot Nematode (*Meloidogyne incognita*) decreased when applying a spray of Si nanoparticles (0.5 and 1 mg L^−1^). The treatment also improved growth and the absorption of essential elements in the plants invaded by the nematodes.

In a similar approach, nanosilica was applied with *Penicillium* sp, an entomopathogenic fungus, to control *Myzus persicae*, a potato (*Solanum tuberosum* L) plant pest that impairs potation production [118]. *Penicillium* sp. can have toxic effects on many insect pests but with loss of efficacy against *Myzus persicae*. Recently, *Penicillium* sp. has been added to micronutrients, including silica, to enhance crop protection from biotic stress. In this sense, nanosilica in combination with *Penicillium* sp. at different concentrations (only nanosilica (1, 3, and 5%) and a mixture of nanosilica and *Penicillium* sp. (1, 3, and 5%) was used against *Myzus persicae* in infested cabbage plants in a greenhouse experiment. After five days of application of the treatments, the mortality of *Myzus persicae* was assessed. Single nanosilica (5%) and the mixture of nanosilica (5%) and *Penicillium* sp. (10^6^ spores/mL) increased *Myzus persicae* mortality by 32.5 and 37.5%, respectively, compared to 12.5% mortality after treatment with only *Penicillium* sp. [118].

Silicon-based NMs can be used with other NPs to alleviate biotic stress in plants. Recently, a silver/silicon dioxide (Ag/SiO_2_) nanocomposite obtained by the biogenic route showed an antifungal effect against *Botrytis cinerea* (chocolate spot disease) in faba bean (*Vicia faba* L.) [119]. The nanocomposite, biosynthesized by the free-cell supernatant of *Escherichia coli* D8, demonstrated antifungal activity with a minimum inhibition concentration (MIC) value of 40 ppm. In vivo studies of infected plants revealed this nanocomposite’s importance in increasing fava bean resistance against *Botrytis cinerea* by increasing the total phenolic content and the activities of antioxidant enzymes (polyphenol oxidase and peroxidase) [119]. The nanocomposite Ag/SiO_2_ had similar effects compared with the positive control (Dithane M-45) in the fungal inhibition (Figure 3).

In addition to nanosilica, selenium NPs (Se NPs) are known as an ecologically and environmentally friendly approach to increase crop production by mitigating (a)biotic stresses since Se activates plant defense mechanisms [120]. Biosynthesized SeNPs have antimicrobial effects against phytopathogens such as fungi and bacteria [121]. SeNPs can generate ROS, which damages the pathogen cell wall, impairing pathogen cell membrane integrity and inhibiting ATP synthetase activity [122]. SeNPs can also directly inhibit pathogen growth by damaging the cell wall and altering deoxyribonucleic acid replication, food metabolism cycle, protein synthesis and modification, thus killing the microorganisms [120].

Another important class of nanomaterials to alleviate biotic stress in plants is carbon-based nanomaterials. Carbon nanotubes (CNTs) are hybridized carbon atoms hexagonally arranged in a laminar structure yielding cylindrical tubes with lengths in the micrometre range and dimensions of a few nanometers. CNTs can be used as antimicrobial agents in plants. For instance, CNTs mitigated the adverse effects of *Alternaria solani* in tomato crops. This disease causes yield losses throughdirect antimicrobial activity and by inducing the plant antioxidant defense system [123]. Indeed, the flavonoid content, ascorbic acid, and glutathione peroxidase activity were improved in plants by the administration of CNTs. The antimicrobial activity of CNTs is based on the generation of ROS after nanomaterial uptake by plant tissue, decreasing the severity of *A. solani*. Interestingly, in plants, ROS production induces the formation of phytohormones related to stress, such as jasmonic acid, salicylic acid, and abscisic acid, in addition to NO, a key player in plant defense [124].

## 7. Conclusions, Challenges and Future Remarks

With the increase in the global population and the need for adequate food production, in light of sustainability, nanotechnology for agricultural applications has emerged as an exciting and promising approach to improve plant growth under (a)biotic stress conditions. Overall, the administration of nanomaterials can significantly improve plant growth, and, in the case of biotic stress, nanomaterials can mitigate the harmful effects caused by phytopathogens. Biotic stress is still a cause of high yield loss in agriculture, with ca. 20–40% of crop yield loss caused by pathogens and pests [112]. Nanomaterials can have a direct toxic effect on plant pathogens and/or can load chemicals that have antimicrobial effects and are sustained [125]. Although much progress has been achieved with the use of nanomaterials in agriculture, important aspects also need to be clarified. NMs have been studied extensively on an experimental or pilot scale. However, many studies are still needed at the scale of hectares of crop fields and at the scale of several years of applications to verify their potential long-term impacts on the soil and its microbiome, plants, and fauna [126,127]. More information is also needed on their safe and profitable commercial application and the safety of foods produced using NMs [128]. The challenges in the use of nanomaterials to combat biotic stress can be highlighted as the need to better and further evaluate nanomaterial uptake, translocation, and modification in plant tissue. The effect of nanomaterials on plants strongly depends on several features, such as the chemical nature of the nanomaterial, size distribution, surface charge and chemical surface, shape, dose, concentration, route of application, duration of treatment, interactions between the nanomaterial and the targeted plant, and environmental microbiota [82]. As expected, at low concentrations, positive effects of the nanomaterial can be observed, whereas toxicity is found at high concentrations/doses. Nanoparticles have been widely studied for their ability to degrade toxic compounds present in the environment, so their application can fulfill more than one function [129]. Importantly, another challenge is the public acceptance of this technology, as the final consumers of plants treated with nanomaterials are humans and/or animals, and the effects of these technologies on human and animal health must therefore be evaluated. In this direction, ecotoxicological studies of nanomaterials are critical, as well as the provision of a regulatory framework for the introduction of nanomaterials in large-scale crop production. Future research aimed at developing the safe use of nanomaterials on a large scale in crop production is welcome.

## Figures and Tables

**Figure 1 antibiotics-12-00338-f001:**
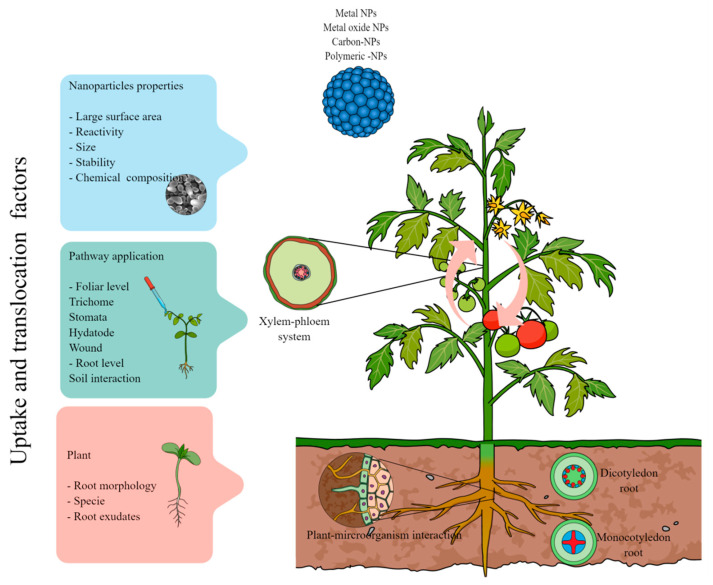
Representation scheme of the main routes used by nanoparticles for translocation in plants.

**Figure 2 antibiotics-12-00338-f002:**
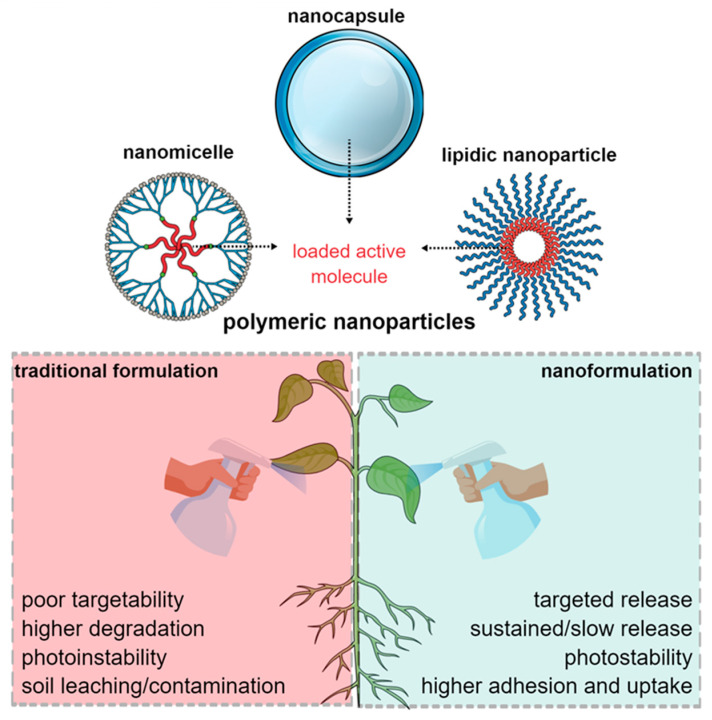
Different types of polymeric NPs used for agricultural delivery applications and their potential against traditional formulations.

**Figure 3 antibiotics-12-00338-f003:**
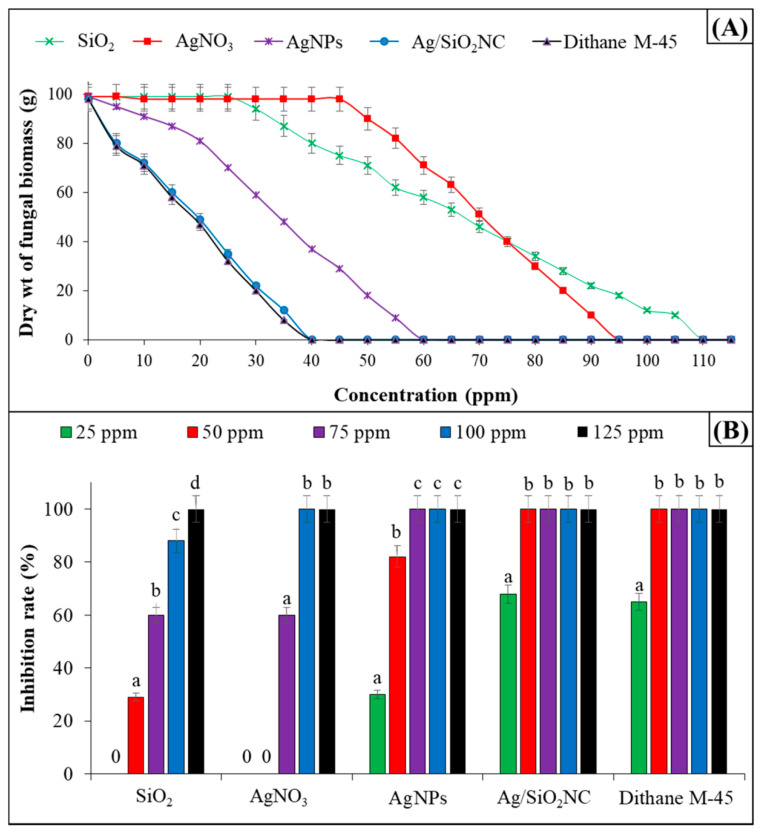
(**A**) The minimal inhibition concentration and (**B**) the inhibition percentage of AgNO_3_, SiO_2_, AgNPs, Ag/SiO_2_ nanocomposite and Dithane M-45 (positive control) against *Botrytis cinerea.* Similar letters (a, b, c, d) are not significantly different at *p* ≤ 0.05 using Tukey–Kramer HSD test. Reproduced from reference [119] under a Creative Commons Attribution 4.0 International License.

**Table 1 antibiotics-12-00338-t001:** Different polymeric nanoparticles and their impact on biotic stress.

NP	Size (nm)	Crop Stress	Impact	Mechanism	Ref.
Chitosan	Not provided	Wilt disease caused by *Fusariumandiyazi* in tomato	In vitro studies showed that, among different tested concentrations (0.1–5.0 mg/mL), 5.0 mg/mL concentration of chitosan NPs produced the maximum inhibition of radial mycelial growth (73.8%).	By inducing the up-regulation of PR-proteins and antioxidant Genes, which play a role in plant defense against pathogen attack.	[70]
Chitosan loaded with paraquat (herbicide)	300	Control of weeds in agriculture	Cytotoxicity and genotoxicity assays showed that the nanoencapsulated herbicide was less toxic than the pure compound.	Lower cytotoxicity and genotoxicity effects of the encapsulated herbicide, compared to its free form, were attributed to the encapsulation effect and the sustained paraquat release.	[73]
Chitosan with and without combination with salicylic acid	Not provided	Rust disease caused by *Puccinia striiformis* (obligate fungal parasite) inoculated in wheat leaf	Infected wheat plants treated with the nanoparticles showed reduction in pustule size and leaf rust when compared to untreated plants.	Increased the activity of antioxidant enzymes, reduction of ROS formation, activation of transcription levels of PR1-PR5 and PR10 genes	[76]
Chitosan loaded with the essential oil peppermint	563	To promote the control of stored food pest for the insets *Sitophilus oryzae* and *Tribolium castaneum*	Significant efficacy of the NPs against both stored product pest compared to control group (untreated)	Inhibition of AChE, which is an essential detoxification enzyme of insect organization.	[77]
Poly(ε-caprolactone) loaded with the herbicide atrazine	483	*Bidens pilosa* (weed species) on soybean plants	Enhancement of herbicide activity and decrease of its toxicity, upon atrazine encapsulation.	Nanoencapsulation of atrazine reduced the levels of applied herbicide applied, due to the sustained release.	[79]
Polyhydroxyalkanoates (PHAs)–of two types– poly-3-hydroxybutyrate [P(3HB)] and poly(3-hydroxybutyrate-co-3-hydroxyvalerate [P(3HB/3HV)] loaded with commercial herbicides	430–750	*Elsholtzia ciliata* weed plants	At the end of the experiment (30 days), the herbicidal activity of encapsulated metribuzin was comparable to the positive control, and all plants were killed. The application of encapsulated herbicides led to the death of weeds, whereas the herbicides remained biologically active, without being prematurely degraded in soil.	Enhancement of herbicide stability upon its encapsulation, which led to a sustained release.	[80]

**Table 2 antibiotics-12-00338-t002:** Different metal-based nanoparticles and their impact on biotic stress.

NP	Size (nm)	Crop Stress	Impact	Mechanism	Ref.
CuO	14–47	*Sitophilus granarius* and *Rhyzopertha dominica* insects that damage wheat grains.	Increased insect mortality by 55–94%; Morphological attributes (lengths, fresh weight, and dry weight of root and shoot, as well as leaves number) and leaf pigments (chlorophylls and carotenoids) were increased.	Stimulating the activity of the enzymes SOD, POD, and APX (antioxidant system) as well as increased concentration of leaf pigments, which have a significant role in scavenging ROS and protecting the plant from stress.	[88]
Ag	23	Bacterial leaf blight (BLB) disease caused by *Xanthomonas oryzae* on rice crops.	Decrease in lesion length of ~31–72% according to Ag NP concentration; decrease in antibacterial activity by 24%; Growth-promoting effect by Ag NPs	Increasing the antioxidant enzyme levels to modulate the adverse effects of reactive oxygen species; promoting nutrient uptake and cellular antioxidative system.	[89]
MgO	20–200	Black shank and black root rot diseases caused by *Phytophthora nicotianae* and *Thielaviopsis basicola*, respectively.	36 and 42% decrease in tobacco black shank and black root rot disease incidence, respectively. Higher inhibitory effect on spore germination, sporangium formation, and hyphal development	Induced ROS production destroys membrane integrity and alters morphological characteristics through pathogen cell uptake. Mg is an essential mineral that participates in numerous physiological and biological processes, playing a crucial role in plant defense.	[90]
TiO_2_	10–100	Yellow stripe rust disease caused by *Puccinia striiformis* on wheat crops.	Inhibition of growth and proliferation of the fungal pathogen resulted in decreased disease incidence and percent disease index when treated TiO_2_ NPs; Promotion of photosynthesis.	Up and downregulation of proteins triggering defense-related responses, such as 6-phosphogluconate dehydrogenase, involved in various reactions of the pentose-phosphate pathways to produce NADPH, which in turn is involved in facilitating the activity of NADPH-oxidase, the main ROS-producing enzyme during infection by pathogens.	[91]
ZnO	13	Fusarium wilt caused by *Fusarium oxysporum* on chickpea crops.	Increase of antioxidant activity and reduction of 90% in disease incidence; Improve photosynthetic rate and fresh and dry weight of roots.	Seed priming with ZnO NPs helped plants accumulate higher quantities of sugars, phenol, total proteins, and activation of defense enzymes such as SOD, PO and CAT, creating resistance against the pathogen.	[92]

## Data Availability

Not applicable.

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
