# Peer review of "Nanoparticles as a Promising Strategy to Mitigate Biotic Stress in Agriculture"

_antibiotics, 2023, doi:10.3390/antibiotics12020338_

Round 1
Reviewer 1 Report
Manuscript ID: antibiotics-2165450
In this review, the authors illustrated the importance of nanoparticles in agriculture. They can be used as nano fertilizers to improve growth and crop productivity by controlling biotic stress. Nanoparticles have also been noted for their exceptional antimicrobial properties. In this regard, metal and metal oxide nanoparticles, polymeric nanoparticles, and others, such as silica nanoparticles, are described in the review. The review is exciting and can be published in the present form.
Author Response
COMMENT: In this review, the authors illustrated the importance of nanoparticles in agriculture. They can be used as nano fertilizers to improve growth and crop productivity by controlling biotic stress. Nanoparticles have also been noted for their exceptional antimicrobial properties. In this regard, metal and metal oxide nanoparticles, polymeric nanoparticles, and others, such as silica nanoparticles, are described in the review. The review is exciting and can be published in the present form.
RESPONSE: Thank you very much for your comments
Reviewer 2 Report
Reviewer Comments1#
Comments for the manuscripts entitled “Nanoparticles as a promissory strategy to mitigate biotic stress in the agriculture” was utilization of various scientific domains an overview of different nanoparticles that have shown the capacity to control biotic stress in plants. Authors have successfully described metal and metal oxide nanoparticles, polymeric nanoparticles and others, such as silica nanoparticles and also covered uptake and translocation. Finally, authors discuss the future remarks about the studies on nanoparticles and their beneficial role in biotic stress management. The review is found to be successful in presenting the appropriate and true knowledge on the ongoing research on nanoparticle based mitigate biotic stress in the agriculture of not fail to sum up and conclude the work. The review is written well and supported with recent reference. But I have some suggestion and recommendations to address the following points before acceptance
1. Author must be discus the advantages and disadvantages of Nanoparticles to mitigate biotic and abiotic stress plants
2. Citation should be added for each statement in the introduction part. Some important and recent reference should be cited in the manuscript, https://doi.org/10.3390/agronomy12092082 https://doi.org/10.1016/B978-0-323-85391-0.00015-0 , https://doi.org/10.1016/j.envres.2022.113821, https://doi.org/10.1016/B978-0-323-85391-0.00011-3, https://doi.org/10.3389/fpls.2022.980046
3. The author should also discuss the one of major issue that the current scenario and toxicity of Nanoparticles and impact towards plat growth and protection
4. Author may also discuss the other smart nanomaterials and their impact on seed germination and immune response in plants.
5. What is the role of nanoparticles against heavy metal stress in plants ? author should also discuss the issue/
6. Author may also discuss the abiotic stress in plants and their impact on stress managements
7. One of the major issues may also discuss, that is role of nanoparticles in pesticides delivery and disease management in plants
8. The author should rewrite the conclusion part for better understanding and explore the current challenges.
The paper should be minor revision and complete characterization and explanation in order before this work can be further considered for publication.
Author Response
REVIEWER 2
Reviewer Comments1#
Comments for the manuscripts entitled "Nanoparticles as a promissory strategy to mitigate biotic stress in the agriculture" was utilization of various scientific domains an overview of different nanoparticles that have shown the capacity to control biotic stress in plants. Authors have successfully described metal and metal oxide nanoparticles, polymeric nanoparticles and others, such as silica nanoparticles and also covered uptake and translocation. Finally, authors discuss the future remarks about the studies on nanoparticles and their beneficial role in biotic stress management. The review is found to be successful in presenting the appropriate and true knowledge on the ongoing research on nanoparticle based mitigate biotic stress in the agriculture of not fail to sum up and conclude the work. The review is written well and supported with recent reference. But I have some suggestion and recommendations to address the following points before acceptance
Reviewer comment 1: Author must be discus the advantages and disadvantages of nanoparticles to mitigate biotic and abiotic stress plants.
Response comment 1: Thank you very much for your comment. The advantages and disadvantages of mitigating abiotic stress in plants are discussed in the manuscript. Moreover, was included a new topic Pag. 5 point 3 "Potential adverse effects of nanoparticles on plants". About abiotic stress, this is out of the Scopus of the review since here we showed the potential use of nanoparticles to combat biotic stress. Moreover, the journal title is "antibiotics", and the special issue is the use of nanoparticles as antimicrobials. Therefore, abiotic stress was not included.
Reviewer comment 2: Citation should be added for each statement in the introduction part. Some important and recent reference should be cited in the manuscript,
https://doi.org/10.3390/agronomy12092082
https://doi.org/10.1016/B978-0-323-85391-0.00015-0
https://doi.org/10.1016/j.envres.2022.113821
https://doi.org/10.1016/B978-0-323-853910.00011
https://doi.org/10.3389/fpls.2022.980046
Response comment 2: Thank you very much for the comments. Some cites were added in page 2. However, others are out of the focus of the review.
Reviewer comment 3: The author should also discuss the one of major issue that the current scenario and toxicity of Nanoparticles and impact towards plat growth and protection.
Response comment 3: Thank you very much for the comment. In the Pag. 5 was included a new topic, "Potential adverse effects of nanoparticles on plants" to treat this theme.
Reviewer comment 4: Author may also discuss the other smart nanomaterials and their impact on seed germination and immune response in plants.
Response comment 4: Nanoparticles' effects on germination and plants' responses are treated in several parts of the review. See Pag 5, 11, 12. The immune responses are out of focus of this review.
Reviewer comment 5: What is the role of nanoparticles against heavy metal stress in plants ? author should also discuss the issue/
Response comment 5: Thank you very much for your comment. However, this topic is out of the Scopus of the review. Therefore, it was not included.
Reviewer comment 6: The author may also discuss the abiotic stress in plants and their impact on stress management.
Response comment 6: Thank you very much for your comment. However, this topic is out of the Scopus of the review. Therefore, it was not included
Reviewer comment 7: One of the major issues may also discuss that is role of nanoparticles in pesticides delivery and disease management in plants
Response comment 7: Thank you very much for the comment. This topic is treated in several parts of the review and new paragrapg were also added. See pag. 10, 11 and 13.
Reviewer comment 8: The author should rewrite the conclusion part for better understanding and explore the current challenges.
Response comment 8:Thank you very much for the comment. The conclusion was enhanced according to the comment.
The paper should be minor revision and complete characterization and explanation in order before this work can be further considered for publication.
Reviewer 3 Report
1.It is suggested to add new research studies about the toxicological concerns. 2.what is the suggestion of this study for future works? 3.Please discuss and compare your results with previous works and add suggestions including Nano antioxidants based methods. 4.It will be better to add the role of nanobiomateriasl and plant stem cell based biomaterials. 5.Please add details for time period and dose selection for endocytosis from previous literatures. 6.More references for the discussion part of manuscript and update and bold your study novelty should be added: e.g.,
-DOI: 10.1016/j.fct.2022.112996
-DOI: 10.3389/fbioe.2022.855136
-DOI: 10.1016/j.jcis.2020.10.047
Author Response
REVIEWER 3
Comments and Suggestions for Authors
Reviewer comment 1: It is suggested to add new research studies about the toxicological concerns.
Response comment 1: Thank you for the comment. A new topic was added in Pag. 5, in point 3: "Potential adverse effects of nanoparticles on plants".
Reviewer comment 2: What is the suggestion of this study for future works?
Response comment 2: Thank you very much for your comments. The suggestions were placed in the item Conclusions, challenges and future remarks.
Reviewer comment 3: Please discuss and compare your results with previous works and add suggestions including Nano antioxidants based methods.
Response comment 3: Thank you very much for the comment. As the present manuscript is not an original contribution, we do not have our own results to compare with previous works since this is a review article (not an original contribution). However, as suggested by the reviewer, we agree that nano-antioxidant-based methods are essential. Indeed, as stated in this review, the main mechanisms of nanomaterials to induce plant defence against biotic stress are the control of ROS generation and the increase of antioxidant defence mechanisms (mainly by the increase of the expression of antioxidant enzymes). This critical feature is already presented and discussed in the revised version of the manuscript, not only in the text but also in Tables 1 and 2 (column 5).
Reviewer comment 4: It will be better to add the role of nanobiomaterials and plant stem cell based biomaterials.
Response comment 4: Thank you very much for the comment. We would like to thank the reviewer for this interesting suggestion. However, this is beyond the scope of this review article, which is based on the recent progress of nanomaterials in plant defense against biotic stress. We present and discuss different kind of nanomaterials. The impact of nanobiomaterials and plant stem cells based biomaterials is out of the scope of this review, and we plan to better discuss this in another review article.
Reviewer comment 5: Please add details for time period and dose selection for endocytosis from previous literatures.
Response comment 5: Unfortunately, time period and dose selection were not found in the literature. Thus, we placed the size of nanoparticle restriction.
Reviewer comment 6: More references for the discussion part of manuscript and update and bold your study novelty should be added: e.g.,
-DOI: 10.1016/j.fct.2022.112996
-DOI: 10.3389/fbioe.2022.855136
-DOI: 10.1016/j.jcis.2020.10.047
Response comment 6: Thank you very much for the comment, the references were added in the pag. 6 and pag. 16
Reviewer 4 Report
This review discusses the importance of nanoparticles and their applications to alleviate the biotic stress in agriculture. Some comments appended below should be considered before publishing in the Antibiotics Journal as follows:
1. The authors should discuss the types of the nanoparticles with brief information about their properties and implementations in various sectors. They could check these recently published articles to support your review (https://doi.org/10.1016/j.arabjc.2017.05.011; https://doi.org/10.1016/j.molliq.2022.121046).
2. The authors are encouraged to incorporate a table, illustrating the applications of polymeric nanoparticles in agriculture, like Table 1.
3. The drawbacks of metal NPs should be presented, which make several biotechnological schools are currently focusing in green synthesis of these NPs. For instance, please check these articles and you could cite them (https://doi.org/10.1016/j.cbi.2022.110166; https://doi.org/10.1038/s41598-022-17712-z).
3. The authors should support this review with other figures, even those published after getting permission from the publisher.
4. The authors should elaborate more about nano silica since it has been broadly applied in the agriculture.
5. Challenge of nanoparticles applications should be highlighted.
Author Response
REVIEWER 4
Comments and Suggestions for Authors
This review discusses the importance of nanoparticles and their applications to alleviate the biotic stress in agriculture. Some comments appended below should be considered before publishing in the Antibiotics Journal as follows:
Reviewer comment 1: The authors should discuss the types of the nanoparticles with brief information about their properties and implementations in various sectors. They could check these recently published articles to support your review (https://doi.org/10.1016/j.arabjc.2017.05.011; https://doi.org/10.1016/j.molliq.2022.121046).
Response comment 1: Information was included about the different types of nanomaterials, their properties, and their use in various sectors.
Reviewer comment 2: The authors are encouraged to incorporate a Table, illustrating the applications of polymeric nanoparticles in agriculture, like Table 1.
Response comment 2: As suggested by the reviewer, we incorporated a table illustrating the applications of polymeric NPs in agriculture (Table 1 in the revised version).
Reviewer comment 3: The drawbacks of metal NPs should be presented, which make several biotechnological schools are currently focusing in green synthesis of these NPs. For instance, please check these articles and you could cite them (https://doi.org/10.1016/j.cbi.2022.110166; https://doi.org/10.1038/s41598-022-17712-z).
Response comment 3: The potential disadvantages of nanomaterials were commented on, and the indicated references were included.
Reviewer comment 4: The authors should support this review with other figures, even those published after getting permission from the publisher.
Response comment 4: As suggested by the reviewer, we inserted a new Figure (Figure 3) with two panels (A, B).
Reviewer comment 5: The authors should elaborate more about nano silica since it has been broadly applied in agriculture.
Response comment 5: Information about the use of nanosilica in agriculture was expanded.
Reviewer comment 6: Challenge of nanoparticles applications should be highlighted.
Response comment 6: We agree with the reviewer, and the challenge of NP applications was highlighted in the last section of the revised version of the manuscript (item 6. Conclusion, challenges and future remarks).
Looking forward to hearing from you, I remain with best regards,
Round 2
Reviewer 4 Report
The authors did a great job to handle the previous comments; thus, I recommend accepting the current version of the manuscript.